# Extent of integration of nutrition assessment counselling and support interventions in the health system and respective drivers: A case of Tororo district, Uganda

Samalie Namukose[1]*, Suzanne N. Kiwanuka[1], Fredrick Edward Makumbi[2], Gakenia Wamuyu Maina[3]

1 Department of Health Policy Planning and Management, School of Public Health, College of Health Sciences, Makerere University, Kampala, Uganda, 2 Department of Epidemiology and Biostatistics, School of Public Health, College of Health Sciences, Makerere University, Kampala, Uganda, 3 Department of Community Health and Behavioural Sciences, School of Public Health, College of Health Sciences, Makerere University, Kampala, Uganda

* snamukoseb@gmail.com

## Abstract

### Background

Uganda embraced Nutrition Assessment Counselling and Support (NACS) since 2009 as a health system strengthening approach to improve health and nutrition outcomes. However, scant evidence exists on NACS integration and drivers. This study therefore assessed the extent of NACS integration in the health system and identified key drivers and barriers.

### Methods

A mixed method design was employed. In a facilitated panel discussions at each of the 17 health facilities, 4–5 health staff participated, responding to a semi-structured questionnaire. Integration was assessed on a 5-point scoring scale of 1 for not done nor integrated, 2–4 for partial and 5 for fully integration. Data was captured, analysed in microsoft excel and presented using as bar and spider charts. Integration drivers were identified deductively from key informant and in-depth interviews using Atlas.ti 9 and thematic analysis.

### Results

The NACS integration across the health facility level was partial at a score of 2.9 indicating a weak integration into the health system. Integration across the health system building blocks was partial at; service delivery (3.8), health work force (3.7), health information (3.3), community support system (3.0), governance and leadership (3.0) signifying that NACS activities are provided by Ministry of Health but sub-optimal due to weak capacities. Health financing (2.2) and Health supplies (1.5) were the least integrated due to partner dependence. Under service delivery, deworming (5) was fully integrated and provided by Ministry of Health. The key drivers for integration were; good leadership, financing, competent staff,

**Data Availability Statement:** All relevant data are within the paper and its Supporting Information files.

**Funding:** The authors received no specific funding for this work.

**Competing interests:** The authors have declared that no competing interests exist.

quality improvement approaches, nutrition talks, community dialogues, nutrition logistics and supplies.

## Conclusion

The NACS integration in the health system was generally low and lacked adequate support. Governance, financing and community follow-up under service delivery require more government investment for enhanced integration.

## Background

Globally, 22% of children under five years of age are stunted, 6.7% wasted, and 14.6% had low birth weight according to Global Nutrition Report 2021 [1]. Additionally, the state of Food Security and Nutrition in the World of 2021 report indicated that 29.9% of women of reproductive age suffered from anaemia while 42% of children under 5 years of age were affected. Overweight and obesity was found to be increasing rapidly nearly in every country in the world. Notably Sub-Saharan Africa and South Asia were identified as contributing the majority of the observed burden of malnutrition [2].

In Uganda, 29% of children under 5 years are stunted, 11% underweight, 4% wasted and 53% anaemic while the prevalence of anaemia and chronic energy deficiency among women of reproductive age was 32% and 12% respectively as reported in the Uganda Demographic Health Survey 2016 [3]. Maternal malnutrition has detrimental effect on the mother and her offspring. Maternal consequences include; low pregnancy weight gain, increased infant and maternal morbidity and mortality, lethargy and reduced productivity. Infant consequences include; intrauterine growth retardation, low birth weight, pre-term birth and reduced intelligence quotient [4,5].

To address the situation in Uganda, the Ministry of Health (MoH) implemented various programs such as the Baby Friendly Health Facility Initiative (BFHI) [6], Infant and Young Child Feeding (IYCF) [7], Integrated Management of Acute Malnutrition (IMAM) [8], Vitamin A Supplementation and Deworming [9]. However, these programs have been implemented in a vertical manner targeting specific population segments, mainly children aged 0-59months. This limitation prompted the adoption of the Nutrition Assessment Counselling and Support (NACS) approach.

The NACS approach, initiated in 2009 in Uganda, aimed to integrate nutrition interventions into policies, programs and health service delivery for all segments of the population across the life cycle. It focused on a more holistic, simple, realistic and innovative ways of integrating and implementing preventive, supportive and curative nutrition services by involving all stakeholders and key sectors for more sustainability [10]. The process started with development of a comprehensive NACS training package with interventions to be provided to clients whenever they accessed health care in selected pilot districts. In these districts, health facilities were provided with; nutrition guidelines, job aides, nutrition supplies and logistics to enhance quality nutrition service delivery. The health workers were trained and empowered to; promote health facility-community linkage for continuum of care, monitor nutrition performance using data capture tools, utilise quality improvement approach to address the performance gaps, and work in partnership with stakeholders to increase coverage of nutrition services.

Regarding service delivery, the NACS intervention to package was provided to all the clients at every visit tailored to their needs. To mothers and their children, support was provided on

optimal maternal nutrition practices which included iron/folic acid supplementation, counselling on a diversified diet and an extra meal, adequate consumption of iodine, deworming, prevention of malaria and counselling on the recommended antenatal visits.

Nutrition education and counselling emphasized optimal breastfeeding practices including early initiation of breastfeeding within the first hour of birth, exclusive breastfeeding for the first 6 months, continued breastfeeding for two years and beyond for non HIV exposed infants and one year for the HIV exposed infants who are confirmed negative on an HIV test. Mothers of older infants >6 months received nutrition education on optimal complementary feeding practices including frequency of meals, adequacy of the foods, thickness of the food, variety of the foods, active feeding, observation of personal, environmental and food hygiene at all processes of food preparation. Mothers/caretakers of sick children were counselled on continued breastfeeding and frequent feeding for infant >6 months [23]. The community system served as continuum of health and nutrition care providing livelihood and economic strengthening support as well as client follow up. The NACS approach aimed to integrate nutrition into health system but the extent of this integration at both the health facility and community levels has not been assessed.

The concept of integration is interpreted differently by various users in the health system and can be defined based on typology, breadth, processes, intensity and health system perspective [11,12]. Several models have been used to describe integrated care depending on the scale and these range from individual, group and disease specific and population based models [11–14]. These studies provide insights on the concept of integration but may not delve deeply into measurement methodologies of integration. For instance, Ellen Nolte et al. [11] and World Health Organization [12] focus on describing integration models while Ahgren et al. [13] and Jason Cheah [14] concentrate on theoretical framework for integration. This paper defines integration from the health system and intensity perspective as the extent to which NACS interventions are assimilated into the health system functions, whether partially or fully [15]. Although service integration has been cited as a strategy for strengthening health systems by reducing service fragmentation and improving access to diverse and interrelated healthcare, it faces challenges in poorly performing or under-resourced health systems. [12,16–19]. This study adapted an integrated nutrition service delivery conceptual framework **S1 Fig** mirroring the Chronic Care model which encompasses many processes and activities performed at the health facility and community level to improve the clinical nutrition outcomes of the beneficiaries.

Regarding benefits, several studies have shown that integrated health care is patient centered, cost effective, increase access to services and results into better health outcomes [14,20–22]. These studies differ in their emphasis on conceptualization and empirical evidence of integration. Atun et al. [15] focuses on the conceptual side while Oxman et al. [21] and Ryman et al. [22] leans towards exploring the practical advantages of integration. A thorough analysis would entail combining both theoretical frameworks and empirical evidence to fully understand the benefits of healthcare integration. Regarding the extent of integration, one study that assessed the level and trend of integration of acute malnutrition management in Niger found variations in integration across the health system functions. Governance and health information scored the highest while financing the lowest. The authors recommended adapting an indicator matrix aligned with national health, nutrition policies and development plans [16]. Deconinck et al. [16] in this study highlights integration process, level of integration and lessons learned. More detailed information on the methodology and deeper examination of the drivers to integration would enhance its impact. Findings from one health system performance assessment on IMAM/NACS conducted in Uganda in 2014 revealed a number of gaps in the health system building blocks. The same authors in this report acknowledged the fact that the

assessment was not meant to be representative of the provision of NACS services in all Uganda, but provide examples of the strengths and gaps in the health system that affect implementation [23]. A systematic review assessing the impact of integrated health and nutrition programs' impact on specific nutrition outcomes indicated that service delivery and health workforce were well integrated but governance, information systems, finance, supplies and technology were less integrated [24]. In comparison to the study by Deconinck et al. [16], Salam et al. [24] offers a broader perspective on integration across health system including nutrition specific and sensitive interventions as well as the enablers and barriers to integration. To gain a deeper understanding of the extent of integration, it is important to assess this concept by level of service delivery and beyond the confines of the health system. It is important to note that no single successful model of integration has been reported. Instead the effectiveness of integration varies according to the context and demands of a particular setting in which it occurs. This variation is evident in a systematic review that showed that integration had a positive impact on nutrition and non-nutrition outcomes with the model of integration adapting to the context and demands of a particular setting [25].

Despite a decade of NACS implementation in Uganda, there is paucity of information on the extent to which NACS has been integration in the health and community system functions. Furthermore, evidence is scarce on the drivers of integration.

Therefore, this study aimed to assess the extent of NACS integration within and beyond the health system and identify key drivers to this process. The study specifically examined the incorporation of NACS within the following domains: 1) healthcare facility levels, 2) health system building blocks and the community system. The findings of this study provides insights into level of assimilation, best practices and areas for improvement which can inform the scale up of NACS initiatives in Uganda.

## Materials and methods

This study was approved by; the Higher Degrees, Research and Ethics Committee–Institutional Review Board, Makerere University School of Public Health (MaKSPH HDREC 24/01/2017), Uganda National Council of Science and Technology (SS 4251) and the Office of the President Uganda (ADM/194/212/01). A formal letter from the Ministry of Health was written to the Tororo District Health Officer to seek for permission to conduct the study. The Principal Investigator informed both the District Health Officer and the District Resident Commissioner about the study plan before its execution. Health workers and participating mothers were asked to sign informed consent form. The mothers who were unable to read and write provided their informed consent using their thumbprint.

### Study setting

Tororo district located in the eastern region was selected due to its prior experience in implementing NACS and a high prevalence of malnutrition reported in the Uganda Demographic Health Survey, 2016 as 30.8% stunting, 3.8% wasting, 14% underweight, 47.8% anemia among children under 5 years of age. Additionally, 17.7% of women of reproductive age were anaemic, and 20% were undernourished (BMI<18.5) [26]. The district has healthcare infrastructure comprising 4 Hospitals (2 government owned, 2 private not for profit), 3 Health Centre IVs (all government owned), 20 Health Centre IIIs (18 government, 2 private not for profit) and 37 Health Centre IIs (35 government—owned, 1 private not for profit, 1 private for profit).

Health Centres I and II constitute primary healthcare facilities, Health Centres IIIs, IVs and Hospitals provide secondary care services, while Regional and National Referral Hospitals offer tertiary healthcare [27].

## Study population and selection

Seventeen (17) health facilities were purposively selected and assessed based on NACS implementation experience and client load. These included; 2 Hospitals, 3 Health Centre IVs and 12 Health Centre IIIs. In addition, 17 facilitated panel discussions, 24 Key Informant Interviews and 22 In-depth Interviews were conducted.

Health facility in-charges, including medical superintendents, senior clinical and nursing officers, and maternal, child health and nutrition staff mainly midwives and nurses participated in the facilitated panel discussions. Key informant interviews involved the District Health Educator, Assistant District Health Educator, Medical Superintendent and Health Facility in charges. In-depth interviews targeted experienced NACS implementation staff. Selection of the study sites and respondents commenced on 7th June 2018, with data collection concluding on 10th July, 2018.

## Study design

A mixed method design was employed to assess the level of integration of NACS in the health system given the intensified NACS support provided to Tororo district from 2012 to 2018. The extent of NACS integration was measured on a 5-point scale guided using a semi-structured questionnaire. The drivers of integration were extracted from the key informant and in-depth interviews of health staff in Tororo district.

## Data collection tools and procedures

**Study instruments.** The semi-structured questionnaire for this study was developed by adapting questions from studies on integration [28,29], benchmarking tools on policies and practices [30] and aligned to national health service standards, nutrition service delivery assessment tool [31,32]. This study used a health systems approach to assess the extent to which NACS activities have been assimilated in each of the health and community system functions and aligned to global and national definition. Governance and Leadership meant existence of strategic policy and legislative framework, effective oversight, provision of incentives and accountability [32]. Health financing meant availability of funding and the right financial incentives that enables individuals have access to effective public health and personal health care [33]. An effective health information system meant production, analysis, dissemination and use of reliable and timely information on the health system performance [34]. The health workforce meant availability of competent and dedicated staff [35]. An efficient service delivery meant physical presence of items required for service delivery for example the infrastructure, qualified staff and the way the governance task determined their use for an efficient system [36]. Health supplies and logistics meant availability or stock out of the nutrition commodities.

We used key informant and in-depth interview guides to collect information on the respective drivers of integration for each health system building block. The interview guides captured information on the experience, facilitating and inhibiting factors to integration.

**Measuring the extent of NACS integration.** The extent of NACS integration was assessed using a facilitated panel discussion of 4–8 participants using a semi-structured questionnaire. Only one tool was filled per facility in which the participants reached consensus on the level of NACS assimilation using prescribed NACS constructs within the health and community systems as outlined in Table 1. The panelists were allocated reference numbers, equal opportunity and minimum time to share their views.

The intensity of integration scores was measured on a scale of 1 to 5 as follows;

1. NACS activities are not integrated and not provided- (No integration)

2. NACS activities are partially integrated and provided by partners (Done by partners / Partial*)

3. NACS activities are partially provided by the Ministry of Health but not fully covering all the needs because of weak capacities–(Weak and not supported/ Partial**)

4. NACS activities are partially provided by Ministry of Health with some support from partners—(Some support from Partners / Partial***)

5. NACS activities are fully integrated and provided by Ministry of Health–(Full integration)

## Data processing and analysis

Microsoft excel was used to analyze the quantitative data. The final scores on the extent of integration from the facilitated panel discussions were recorded in an excel spread sheet per health facility categorized according to health system function. The sum scores of all health facilities at the same level was computed to determine the mean (standard deviations) integrated score for each health system function. The mean integration scores were presented using spider and bar charts to depict the extent of integration of NACS; 1) across the health facility levels, 2) into health building blocks extending into the community system.

ATLAS.ti 9 was used for the qualitative data analysis to complement quantitative findings. Data from the audio recorded key informant and in-depth interviews were transcribed and coded based on the identified themes from the research questions. The sub themes and themes were then categorised and aligned to each of the health system functions. Thematic analysis was employed and relevant quotes were used to illustrate key points.

## Results

### NACS integration across health facility level

The overall integration of NACS across the health facility level is presented in S2 Fig. The mean integration scores for NACS across the health facility level was 2.9, indicating that NACS activities are weak and mainly supported by partners (Partial*). Hospitals had higher integration scores at 3.0 compared to Health Centre IIIs (2.9) and Health Centre IVs (2.8).

### Integration of NACs across health systems building blocks

The extent of NACS integration in each of the health system building blocks is illustrated in S3 Fig.

The service delivery function scored 3.8, followed by health work force (3.7), health information (3.3), and Governance and leadership (3.0). This indicates that that NACS activities are provided by the Ministry of Health but not fully covering all the needs because of weak capacities (Partial**). Health financing and health supplies functions were the least integrated with mean integration score of 2.2 and 1.5 respectively. This meant that NACS activities are mainly primarily provided by partners through vertical programming.

**Governance and leadership.** Integration of NACS in the governance and leadership function was slightly higher in Hospitals at 3.2 compared to Health Centre IVs (3.1) and Health Centre IIIs (2.9). This meant that NACS integration is partial** whereby activities are provided by the Ministry of Health but sub-optimal due to weak capacities.

In addition, participants mentioned that working as a team, having regular supervision, ensuring necessary knowledge and skills and regular were key ingredients of good leadership as alluded to in the following quote:

**Table 1. Constructs used to measure the extent of integration of NACS in the health system.**

| Health System Building blocks | Constructs considered |
|---|---|
| Governance and Leadership | • Availability of integrated Policies, Standards and Guidelines,<br>• Availability of integrated Technical Leadership and co-ordination<br>• Availability of integrated Comprehensive workplan and budget |
| Health/Nutrition Financing | • Pooled funds from the public and local partner sources<br>• Nutrition included in the Health facility workplan<br>• Workplan transparent and understandable to the service providers, users and the public<br>• Staff involved on the NACS on the pay roll |
| Health/Nutrition information | • Nutrition indicators integrated in the Health Management Information System<br>• Availability of Registers with nutrition data elements<br>• Staff trained in Health Management Information System (HMIS)<br>• Key NACS performance indicators monitored, reported, graphed and displayed |
| Health/Nutrition Workforce | • Availability of Human resource trained in NACS<br>• Dedicated staff for nutrition<br>• Availability of clear job description with NACs components captured<br>• In-service training/Continuing Professional Development (CPD)<br>• Performance appraisals with NACs integrated<br>• Motivation systems in place |
| Health/Nutrition supplies and Logistics | • Availability of integrated Supply chain management,<br>• Nutrition commodities forecast and procured<br>• Availability of stock cards reflecting nutrition commodities<br>• Therapeutic foods managed as essential drugs and supplies |
| Nutrition Service delivery | • Health, Nutrition Education and Counselling<br>• Nutrition Assessment and Categorization for the mothers and infants<br>• Provision of appropriate support to the malnourished mothers and infants<br>• Micronutrient supplementation and deworming to infants and mothers<br>Active follow up of the mothers and infants including linkage of the mother-infant pairs to the Community for continuum of health and nutrition care.<br>• Quality Improvement Projects on nutrition<br>• Capacity building activities; training, Continuing Professional Developments, coaching, mentorships and supervision |
| Community Support System | • NACS related activities integrated in the community outreaches<br>• Availability of Community based health workers attached to the health facility<br>• Clients referred to economic/ livelihood support by a named service provider<br>• Availability of integrated functional follow up mechanism by the facility |

*. . .actually when you compare this facility to other facilities, our leadership is strong and welcoming new ideas. When something is brought on board, it is hard to be dropped because of team work, supervision, the follow up. Are you doing this, are you doing that! Remember it is like this, so something like that, it is backed up by continous mentorship, reporting, if something is done and you are not seeing where it is being reported, you are likely to drop it . . . (IDI_HW_ OSUKURU HCIII)*

**Health financing.** NACS integration was partial* (activities were mainly partner supported) in the health financing function with an average score of 2.2. However, Health Centre IVs and Health Centre IIs scored 2.3 while Hospitals scored 2.0.

Furthermore, participants commonly mentioned that provision of financial and material support including procurement of anthropometric equipment and therapeutic food, capacity building activities, nutrition assessment follow up of malnourished cases, quality improvement activities, establishment of food demonstration gardens, and printing of data collection tools played a crucial role in facilitating nutrition service delivery. Interestingly, one participant recommended that partners should shift their focus towards supporting districts in strategic

planning and interventions that have long lasting impact rather than providing short term. This approach was seen as more sustainable as illustrated in the following quote:

> *The partners should support districts to plan strategically and intervene in strategic areas which can have a long-lasting impact other than thinking of giving handouts, it is not sustainable to give handouts. Handouts can be given in form of therapeutic food for those who are severely malnourished,. . . (KII_MS_TORORO HOSPITAL)*

**Health workforce.** The integration of NACS was sub-optimal in the health workforce function with a mean score of 3.7. Health Centre IVs scored higher at 4.3, compared to Health Centre IIIs at 3.7 and Hospitals at 3.0.

In addition, participants highlighted the importance of having adequate and competent human resources as a driver for NACS integration. They explained that staff who were passionate, motivated, skilled, committed, and possessed a positive attitude and team spirit were instrumental in mainstreaming NACS interventions in health service delivery, as illustrated by the following quote;

> *. . .. the availability of the human resource. . .and then team work. Team work is very key. Even if you have the personnel but if you are not able to work as a team, you may not achieve much. Then secondly we have also been able to empower the patients such that in case this has not been done they are able to say that my hand has not been measured (KII_MUKUJJU HCIV)*

Furthermore, capacity building activities emerged as another key driver to NACS integration. Participants mentioned that training health workers, conducting regular continuous professional development sessions, and providing regular supervision enhanced the knowledge and skills of the health workers and was a constant reminder to them to stay focused and adhere to the nutrition guidelines and standards.

> *. . ..we also have regular CMEs (Continuing Medical Education). They also help to remind people about certain programs. If you do not do it, people will just forget and go for other programs. (KII_MELLA HCIII)*

Additionally, the participants appreciated the role that the support staff such as Linkage Facilitators and Village Health teams played in the provision of nutrition services especially nutrition assessment. The staffing levels of the health facilities was far below standard (<50%) to enable the health workers to conduct nutrition assessment for every client and this is illustrated by the following quote:

> *. . ...those linkages are always available; you know the staffing levels currently in this facility we are like at 50%, and then you are like telling us; MUAC for every client, you know! The provision of the support staff has really helped us a lot (IDI2_ HW2_POYAMERI HCIII)*

From the demand side, the participants mentioned patient empowerment as a key driver to integration. They reported that the patient's rights and responsibilities are pinned on the notice board as a source of information to clients on services provided and what is expected of them. Their knowledge is enhanced through health and nutrition education talks as well as nutrition counselling. As a result, the clients always demanded for the services whenever the health workers had missed out any. The following quote highlights these assertions;

*We start with a health talk, and when you check our notice board we have the patients' rights, those are one of the things spoken about on a daily basis; the patients' rights and responsibilities and what they should expect when they come to the health facility. And if it is not done then they need to demand (KII_MUKUJJU HCIV)*

**Health information.**    NACS integration score into the health information function was partial** with a score of 3.3. Hospitals and Health Centre IIIs had better integrated nutrition information with a score of 3.6 and 3.7 compared to Health Centre IVs which scored 2.7.

Additionally, participants pointed out that the availability of data collection, reporting tools and proper documentation were key drivers to NACS integration. They mentioned that there were deliberate efforts to track nutrition integration by reviewing their registers. The regular internal and external support supervision thrusted the health workers to improve on their documentation as supervisors always demanded to review the registers and address the gaps on spot as reflected in the following quote:

*Those mentorships, and support supervision, they keep coming and when they come they ask for those registers, when they find that you are not doing it well, they take you through it. And the opening of those journals, you open a journal, you have started at this percentage and you want to achieve this you keep on pushing and somehow they embrace it (KII_OSUKURU HCIII)*

The quality improvement approaches including performance reviews employed were mentioned as drivers of NACS integration. These ranged from problem analysis to identify gaps and solutions, opening of documentation journals for nutrition projects, use of dash boards to monitor performance, mentorships, learning sessions to share experience and harvest meetings to gather best practices that could be replicated in other health facilities.

*The mentorships; structured mentorships were programmed where achievements were aimed at different levels, at different time. So we had a QI project on nutrition and we were monitoring and learning sessions were arranged, so members would go, come back implement and review the performance. So the habit of reviewing performance routinely and dashboard really kept us on our toes (KII_MS_TORORO HOSPITAL)*

**Health supplies.**    The health supplies function was the worst integrated with a mean score of 1.5 across the 3 health facility levels meaning activities are majorly partner supported and vertical. Hospitals had higher scores at 2.5 compared to both Health Centre IVs and Health Centre IIIs at 1.0.

Furthermore, participants mentioned that availability of anthropometric equipment, therapeutic foods and data collection tools as drivers to NACS integration. The supplies which ranged from weighing scales, MUAC tapes, height boards, registers and reporting tools were procured and distributed by partners as illustrated from the following quote.

*. . .. implementing partners when they come for their assessment, they find that some of us are not trained in NACS, other tools like weighing scales were not there, they had to buy for us. Then TASO brought for us the MUAC tapes, registers we didn't have them, these were brought (KII_MELLA HCIII)*

**NACS integration in health service delivery.**    Results of a detailed analysis of the extent of integration in the health service delivery function are presented S4 Fig.

Deworming had the highest integration score with a mean score of 5.0 indicating full integration and provision by the Ministry of Health. Integration score of 4.9 for micronutrient supplementation and 4.1 for nutrition assessment meant that these activities are provided by the Ministry of Health with some support from partners. The integration score of 3.9 and 3.8 for health and nutrition education and quality improvement respectively meant that the activities are provided by the Ministry of Health but not fully covering all the needs because of weak capacities. Capacity building activities (2.9) and follow up care (2.4) had the least integration scores meaning the activities are not integrated but provided by partners.

Participants mentioned health and nutrition education talks as key driver to integration, explaining that they served as a source of information dispelling myths and misconceptions about malnutrition. Additionally, food demonstrations empowered clients to put into practice what they learnt from the health facilities.

*So what we did, we could give health education and also with support from implementing partners like TASO, and world vision, they could bring the food and facilitate that activity so it really helped so much (IDI-2_HW_ KIYEYI HCIII)*

**Integration in community support system.** The results of the NACS integration in the Community system are shown in S5 Fig.

Integration scores of NACS in the community support system across the 3 health facility levels was partial** with a score of 3.0. Health Centre IIIs had slightly higher integration at 3.3 compared to Hospitals (3.0) and Health Centre IVs (2.7).

Most importantly, the participants emphasized the significance of the community arm in integration since it provided a continuum of health and nutrition care to clients. Furthermore, they explained that community dialogues had a transformative impact towards adoption of optimal nutrition practices, consequently reducing malnutrition in the community as illustrated in the following quote.

*. . . before RHITES-E came, they used to have community dialogues, people could go, get a topic . . .and teach about Nutrition and emphasize the need to change. So that has also made malnutrition to reduce in the community (KII_ KWAPA HCIII)*

## Discussion

This study assessed the extent to which NACS elements were assimilated in the health system functions while also identifying the key factors driving the integration. The findings serve as valuable insights for the potential scale up of NACS in Uganda. The overall, integration scores for NACS across health facilities and health system building blocks and their implication for nutrition outcomes are discussed. Our deep dive at NACS integration at service delivery level revealed areas of weakness which further compromise nutrition outcomes. Further, this paper highlights the important role of community systems which are vital to the success of nutrition programs.

The findings showed that NACS was mostly integrated in the service delivery function with a mean score of 3.8. That is, the activities were provided by MoH with some support from partners. However, follow up care was weakest although lower health facilities tended to perform slightly better than the higher level facilities. This could be justified by the fact that they tend to be closer to communities. Similar to several studies, integration in the service delivery function scored highest because most programs are able to offer integrated services through the existing

delivery mechanisms [24,25,28]. Consequently, community support systems such as integrated community outreaches, follow up and referral mechanisms, program specific community resource persons and frequent supervision need to be strengthened for better nutrition outcomes. This is supported by Emma Sacks et al.[37] who advocate for an expansion beyond the WHO building blocks and the allocation of technical and financial resources to community health to enable provision of primary health care for all.

The integration of NACS in the governance and leadership block was partial with a mean score of 3. This means that the NACS activities are being integrated but are compromised by weak capacities in terms of availability of the necessary protocols, planning, budgets and coordination which should be provided by the district health office and Ministry of Health. On the other hand, good leadership was the driving factor for the integration of nutrition interventions into health policies, plans, stakeholder coordination and ensuring sustainability of the interventions. These findings are comparable to several studies that suggest multi-sectoral coordination, leadership and planning as enablers to integration [20,24,38]. Unfortunately, this assessment did not cover the national level structures and processes so as to complete the loop from national level planning to implementation and may therefore be unable to elucidate why these gaps persist.

Across the 3 health facility levels, integration scored second lowest on health system financing with a mean score of 2.2. This may be an indicator that the financing of nutrition interventions heavily depends on donor support. The lack of pooled budgets and health workplans reflecting nutrition activities highlights the lack of investment in nutrition ultimately constraining nutrition service delivery. Although partner support was highlighted out as one of the drivers to integration, it was noted that there was no transparency, co-ordination of partner funding, commonly agreed upon funding allocation and poor integration in the district planning cycles. Although partner support facilitated integration, it was observed that this support might be unsustainable. Several studies have revealed similar challenges of sustainability of interventions due to donor driven projects [39–42]. This implies that sustained implementation of nutrition interventions requires reasonable financial contribution from government.

The integration of NACS into the human resource function was partial with a mean score of 3.7. This implies that nutrition activities are provided by the Ministry of Health but are limited in terms of dedicated staff for nutrition, clear job descriptions, performance appraisals, staff competence, motivation and minimal supervision to lower level health facilities. Motivated and skilled staff, capacity building activities, availability of support staff, and patient empowerment were highlighted as key drivers for integration. Several studies on integration have documented human resource challenges and recommended that adequate and skilled health work force are needed for effective implementation of policies and programs designed by the national governments [17,19,43,44]. Current literature also recommends the recruitment of nutritionists at the various levels with clear job descriptions, the provision of capacity building activities to address service delivery gaps and the motivation of staff through performance appraisals, staff development and local incentives.

The integration of NACS within the health information system function was partial with a mean score of 3.2. This means that services are provided by the Ministry of Health but limited in terms of availability of updated data capture and reporting tools, staff competence in use of the Health Management Information System (HMIS) and District Health Information System 2(DHIS2) as well as poor documentation. In addition, previous studies have highlighted similar challenges of staff capacity gaps, user access to the information systems, poor documentation [45–47]. These weaknesses revealed in the current health management information system imply limitations in tracking progress towards the desired set nutrition targets, underestimation of the burden of malnutrition in the district thus failure to respond appropriately.

The health technologies and supplies function scored lowest with a mean integration score of 1.5. This means that the procurement and distribution of nutrition commodities and supplies is heavily dependent on donor support. Although an initial step has been taken to include the nutrition commodities on the essential medicine list, the majority of the supply chain management functions are not implemented by the Ministry of Health. For instance, the product selection, forecasting, quality assurance, procurement, inventory management and distribution are still solely done by partners. Further reviews of the supply chain system, generated recommendations which were limited to strengthening the supply chain system in the private sector and not the public sector [48]. This implies that sustainable and universal access to nutrition commodities (consumables and non-consumables) will require strategic shift from donor driven support to government led stewardship and coordination of this function.

## Strengths and limitations

In this study, a mixed method design was used to assess the extent of integration, with key health managers serving as the information source which was a strength. However it is important to note that the assessment was limited to a single district which limits generalizability of the findings, representing a weakness.

## Conclusion

NACS integration within and beyond the health system was generally weak and lacked adequate support. Among the building blocks, service delivery showed relatively fairer integration whereas financing and health supplies were notably weaker. Governance, financing and community follow-up under service delivery require more government investment for enhanced integration. The key drivers for integration were; good leadership, funding, competent staff, quality improvement approaches, nutrition talks, community dialogues, health and nutrition education talks as well as nutrition logistics and supplies.

These findings highlight opportunities for improvement in achieving a fully integrated nutrition delivery and the scaling up of the NACS.

## Recommendations

In view of these findings, we suggest the following recommendations to enhance practices and policies related to integration of nutrition interventions and potential areas for future research. In order to strengthen the information system, the Ministry of Health will need to address gaps across the system by providing updated registers that capture nutrition data elements and indicators to the districts, building the capacity of health staff in data management. Addressing gaps in service delivery will require; building the capacity of a critical mass of services providers in nutrition packages, quality improvement approaches, strengthening the follow up and referral mechanism, utilising the Village Health Teams and linkage facilitators for basic nutrition services.

Future nutrition strategic plans for both national and district level should be developed premised on the six health system building blocks with a solid framework for integration. To enhance ownership, co-ordination, accountability and sustainability of nutrition interventions, the Ministry of Health should empower the districts in governance and leadership skills. Strengthening the health supply chain system will require strengthening the governance and leadership structures of the Ministry of Health to co-ordinate all the key stakeholders in the supply system, building capacity of service providers in the Supply Chain Management functions, allocating central government funds for nutrition commodities, promoting local production, and close monitoring and supervision of the processes. Additionally, we recommend

increased government budget allocation to nutrition at district level, and support districts in leveraging and pooling partner support to a centralized district account for transparency, improved co-ordination and efficient and effective use of resources.

Finally future research could concentrate on assessing the extent of integration of nutrition interventions across all levels of service delivery, including multiple districts and at the national level.

## Supporting information

**S1 Checklist.**
(DOCX)

**S1 Data. Data health facilities Tororo.**
(XLSX)

**S1 Fig. A conceptual framework on NACS integration in the health system.**
(TIFF)

**S2 Fig. Overall integration scores across health facility level.**
(TIFF)

**S3 Fig. NACS integration across the health system building blocks by health facility level.**
(TIFF)

**S4 Fig. NACS integration in service delivery by health facility level.**
(TIFF)

**S5 Fig. NACS integration in community system by health facility level.**
(TIFF)

## Acknowledgments

We appreciate the Tororo district political and administrative team for their permission and support during the data collection period. We thank the supervisors, data collectors, health staff, mothers and their infants for their participation in the study. Finally, Special appreciation go to Drs. Richard Kajura, Godfrey Bwire and Arthur Bagonza for their encouragement and peer support.

## Author Contributions

**Conceptualization:** Samalie Namukose, Suzanne N. Kiwanuka, Fredrick Edward Makumbi, Gakenia Wamuyu Maina.

**Data curation:** Samalie Namukose, Suzanne N. Kiwanuka, Fredrick Edward Makumbi.

**Formal analysis:** Samalie Namukose, Suzanne N. Kiwanuka, Fredrick Edward Makumbi, Gakenia Wamuyu Maina.

**Funding acquisition:** Samalie Namukose.

**Investigation:** Samalie Namukose.

**Methodology:** Samalie Namukose, Suzanne N. Kiwanuka.

**Project administration:** Samalie Namukose.

**Resources:** Samalie Namukose.

**Software:** Samalie Namukose.

**Supervision:** Samalie Namukose, Suzanne N. Kiwanuka, Fredrick Edward Makumbi, Gakenia Wamuyu Maina.

**Validation:** Samalie Namukose, Suzanne N. Kiwanuka, Fredrick Edward Makumbi, Gakenia Wamuyu Maina.

**Visualization:** Samalie Namukose, Suzanne N. Kiwanuka, Fredrick Edward Makumbi, Gakenia Wamuyu Maina.

**Writing – original draft:** Samalie Namukose.

**Writing – review & editing:** Samalie Namukose, Suzanne N. Kiwanuka, Fredrick Edward Makumbi, Gakenia Wamuyu Maina.

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
