## [Decision Letter · Decision Letter 0]

29 Aug 2023

PONE-D-23-21035Extent of integration of nutrition assessment counselling and support interventions in the health system and respective drivers: A case study of Tororo district, UgandaPLOS ONE

Dear Dr. Namukose,

Thank you for submitting your manuscript to PLOS ONE. After careful consideration, we feel that it has merit but does not fully meet PLOS ONE’s publication criteria as it currently stands. Therefore, we invite you to submit a revised version of the manuscript that addresses the points raised during the review process.

Please see the reviewers' comments appended below.

We look forward to receiving your revised manuscript.

Kind regards,

Dr Imran Naeem

Academic Editor

PLOS ONE

4. We notice that your supplementary figures are uploaded with the file type 'Figure'. Please amend the file type to 'Supporting Information'. Please ensure that each Supporting Information file has a legend listed in the manuscript after the references list.

Reviewers' comments:

Reviewer's Responses to Questions

**Comments to the Author**

1. Is the manuscript technically sound, and do the data support the conclusions?

Reviewer #1: Partly

Reviewer #2: Yes

2. Has the statistical analysis been performed appropriately and rigorously? 

Reviewer #1: Yes

Reviewer #2: N/A

3. Have the authors made all data underlying the findings in their manuscript fully available?

Reviewer #1: Yes

Reviewer #2: Yes

4. Is the manuscript presented in an intelligible fashion and written in standard English?

Reviewer #1: Yes

Reviewer #2: Yes

5. Review Comments to the Author

Reviewer #1: Journal Name: PLOS ONE

Title of paper: Extent of Integration of nutrition assessment counselling and support interventions in the health system and respective drivers: A case study of Tororo District, Uganda

Abstract: the abstract satisfactorily captures the gap that warranted the study, the methods used in data collection, the findings, and conclusion. Authors should however consider including the objectives of the study in the background section of the abstract.

Also, which category of health staff were included in the panel discussions? This should be briefly mentioned in the methods section of the abstract, together with the categories of health facilities (primary, secondary, tertiary, etc). This will give a better birds eye view of the entire study in the abstract.

Introduction and Objectives/Research Questions: In lines 45 to 52 of the background, please indicate the period for which the statistics provided were valid. For instance, “The prevalence of anemia among women of reproductive age was 29.9% while that of children under 5 years of age was 42%”. Which year or period was the prevalence 29.9% and 42%? Please indicate and rectify this throughout the study.

Further, please indicate which segments of the population the implementation of BFHI, IYCF, IMAM were limited to. Also, provide references for these initiatives (policies) including the NACS in the writeup and references (citations) for how they were implemented, if possible. This should allow interested readers to follow up on the implementation.

In the literature review from line 97 onwards, kindly consider briefly including the results of the studies reviewed, rather than just indicating what the studies sought to do. Also, the write-up seems to just report the available studies in the field. Kindly consider critically analyzing the studies presented (in a few words, compare, contrast, and critique) to situate the current work in the existing literature.

Further, with the objectives of the study (lines 120-122), service delivery is also a health system building block. As such, objectives 2 and 3 can be merged. Kindly consider doing so, or listing out the specific building blocks that were captured by the study.

Methods: in the study setting, please provide the full meaning of HCIV, HCIII, HCII before using the abbreviations subsequently. Also, which level of service delivery are the hospitals in the district which were captured in the study (primary, secondary, tertiary, etc)? This will help provide some context for the results and discussion.

Please, check “panel facilitated discussions” against “facilitated panel discussions” (it appears the latter is the right term) in line 141.

In line 154, did you mean a 5-point scale?

The section titled “NACS intervention package” in the methods section seems a bit out of place. Was the intervention package provided by the researchers or were they routine activities provided by the health workers? If the latter is the case, then that section should be moved to the background, because it doesn’t seem to serve any purpose in the methods. If the former is the case, then please explain why the need for the intervention in relation to the objectives of the study. The study does not employ a quasi-experimental or action research design. The study just seeks to explore the extent of NACS integration.

Line 184 indicates that the study is a case study. Prior to that, however, the authors just indicate that they used a mixed methods design. The authors should please be specific about the research design they employed. Is the study a mixed methods case study research? Please indicate and let that run through the write-up.

Also, please indicate how the questions on the interview guide were derived. Or were they derived from the same sources as the questionnaire? If so, please indicate (lie 197).

The write-up for the data analysis section is not clear. Please rephrase to show that Microsoft Excel was used to analyze quantitative data while ATLAS.ti 9 was used to analyze qualitative data.

Findings: findings are satisfactorily reported. However, consider indicating the meanings of the various asterisks (* and **) as footnotes for easy reading (provided it is in line with editorial policy of the journal).

Results are satisfactorily discussed.

Conclusion and policy implications: Authors should consider including a brief summary of the entire research before highlighting the key findings and some implications for health policy

Weaknesses: generalizability of the study may not be possible because it was conducted in just one district of the country (Uganda).

Secondly, the authors need to indicate how they controlled for the weaknesses of facilitated panel discussions such as ineffective moderating, dominating panelists, and ill-prepared panelists.

Finally, please consider proofreading and correcting minor grammatical errors throughout the entire write-up.

Reviewer #2: Thanks very much for providing me with the opportunity to review this manuscript titled.“The extent of integration of nutrition assessment counseling and support interventions in the health system and respective drivers: A case study of Tororo district, Uganda Samalie Namukose1, Suzanne N Kiwanuka, Fredrick Edward Makumbi, Gakenia Wamuyu Maina”. It is a well-written manuscript however, we may consider the following to improve the quality of this very informative paper.

Background.

1. (Pg.5, line 119-123) …… There is a need to be very clear on the specific objectives. It appears to me that, there is a repetition (Mentioning health system building blocks and again service delivery; I think service delivery is part of the health system building blocks…and even health facility is a component within service delivery) …… We may consider our specific objective to be 1) To determine the extent of NACS integration across the pillars of the health system building blocks, ii) To determine the extent of NACS integration beyond the health system building blocks (Community health system) ----Detail provided in https://gh.bmj.com/content/3/Suppl_3/e001384 and; 3) To explore facilitators and barriers for NACS integration.

Methodology

2. In the title and abstract case study is mentioned; but in the methodology part case study is not mentioned. Could you clarify?

Discussion

3. It would be better to include the strengths and limitation of the study

4. P.g 21(line 469-487), please reorganize your recommendations into practice, policy level, and research (I think there may be some

6. PLOS authors have the option to publish the peer review history of their article (what does this mean?). If published, this will include your full peer review and any attached files.

Reviewer #1: No

Reviewer #2: **Yes: **James Tumaini Kengia

---

## [Author Response · Author response to Decision Letter 0]

5 Oct 2023

Response to the Academic Editor’s comments

1. Please ensure that your manuscript meets PLOS ONE's style requirements, including those for file naming. The PLOS ONE template can be found at PLOS ONE ; 

The manuscript has been revised to meet the PLOS ONE style requirements and this is reflected in track changes throughout the manuscript

All relevant data are within the manuscript and its supporting information files. We will need help from the Journal in making the data available through publication of the supporting information file as provided.

The ethics statement has been moved to the methods section on page 7 in the revised manuscript with track changes. 

4. We notice that your supplementary figures are uploaded with the file type 'Figure'. Please amend the file type to 'Supporting Information'. Please ensure that each Supporting Information file has a legend listed in the manuscript after the references list.

The file type name has been amended and uploaded with the right file type to ‘Supporting Information’.

Response to Reviewer one’s Comments

5. Abstract: The abstract satisfactorily captures the gap that warranted the study, the methods used in data collection, the findings, and conclusion. Authors should however consider including the objectives of the study in the background section of the abstract.

The objectives for conducting the study has been inserted in the background section of the abstract on page 1 and the statement reads as follows, “This study assessed the extent of NACS integration in the health system and identified key drivers and barriers”.

6. Also, which category of health staff were included in the panel discussions? This should be briefly mentioned in the methods section of the abstract, together with the categories of health facilities (primary, secondary, tertiary, etc). This will give a better birds eye view of the entire study in the abstract.

The health staff ranged from health facility in-charges who were mainly medical superintendents, senior clinical and nursing officers. The maternal Child Health and Nutrition staff included midwives, nutritionists and nurses. Due to word limitation in the abstract, a high level terminology of “health facility staff” has been used for the abstract but expounded in the main body in the methods section to include these cadres.

The categorization of the health facilities has been included in the methods sections on page 7 to read, “Health Centres I and II constitute primary healthcare facilities, Health Centres IIIs, IVs and Hospitals provide secondary care services, while Regional and National Referral Hospitals offer tertiary healthcare”.

7. Introduction and Objectives/Research Questions: In lines 45 to 52 of the background, please indicate the period for which the statistics provided were valid. For instance, “The prevalence of anaemia among women of reproductive age was 29.9% while that of children under 5 years of age was 42%”. Which year or period was the prevalence 29.9% and 42%? Please indicate and rectify this throughout the study.

The wording “Uganda demographic health survey 2016” has been inserted as a reference period for the statistics provided through the manuscript. 

8. Further, please indicate which segments of the population the implementation of BFHI, IYCF, IMAM were limited to. Also, provide references for these initiatives (policies) including the NACS in the write up and references (citations) for how they were implemented, if possible. This should allow interested readers to follow up on the implementation.

The segment of the population has been indicated in the manuscript and these are mainly children, 0-59 moths of age. The references for the various nutrition programs have also been inserted

9. In the literature review from line 97 onwards, kindly consider briefly including the results of the studies reviewed, rather than just indicating what the studies sought to do. Also, the write-up seems to just report the available studies in the field. Kindly consider critically analysing the studies presented (in a few words, compare, contrast, and critique) to situate the current work in the existing literature.

Further, with the objectives of the study (lines 120-122), service delivery is also a health system building block. As such, objectives 2 and 3 can be merged. Kindly consider doing so, or listing out the specific building blocks that were captured by the study.

The studies have been critically analysed to situate the current work in this existing literature as indicated on pages 4-6.

This section has been rewritten to reflect one overall objective of the study. The sub-components to be examined have also been rephrased to address the concern raised. We have also used the additional information you shared in beefing up this paper.

10. Methods: in the study setting, please provide the full meaning of HCIV, HCIII, HCII before using the abbreviations subsequently. Also, which level of service delivery are the hospitals in the district which were captured in the study (primary, secondary, tertiary, etc)? This will help provide some context for the results and discussion.

Please, check “panel facilitated discussions” against “facilitated panel discussions” (it appears the latter is the right term) in line 141.

The abbreviations HCIV, HCIII, HCII have been written in full at their first use on page 7. In addition, the “primary", “secondary”, “tertiary”, level of service has been inserted in the methods section to improve meaning. 

This has been noted, the paraphrase, “facilitated panel discussions” is correct one and this has been addressed.

11. In line 154, did you mean a 5-point scale?

The section titled “NACS intervention package” in the methods section seems a bit out of place. Was the intervention package provided by the researchers or were they routine activities provided by the health workers? If the latter is the case, then that section should be moved to the background, because it doesn’t seem to serve any purpose in the methods. If the former is the case, then please explain why the need for the intervention in relation to the objectives of the study. The study does not employ a quasi-experimental or action research design. The study just seeks to explore the extent of NACS integration.

Thanks for pointing this out. The wording “5-point” has been edited to read “5-point scale”. 

We agree. The wording “NACS intervention packages” has been shifted to the background section on page 4 to enrich the context for the reader. 

12. Line 184 indicates that the study is a case study. Prior to that, however, the authors just indicate that they used a mixed methods design. The authors should please be specific about the research design they employed. Is the study a mixed methods case study research? Please indicate and let that run through the write-up.

Also, please indicate how the questions on the interview guide were derived. Or were they derived from the same sources as the questionnaire? If so, please indicate (lie 197).

Thanks for pointing this out. The study used a mixed method study design. The wording “Case study” has been deleted. The wording “A case of Tororo district” has been maintained in the title for emphasis. 

The interviews were meant to complement the quantitative data. A statement has been added in the methods section page 9 to read, “The interview guides captured information on the experience, facilitating and inhibiting factors to integration”. 

13. The write-up for the data analysis section is not clear. Please rephrase to show that Microsoft Excel was used to analyse quantitative data while ATLAS.ti 9 was used to analyse qualitative data.

The write up for the data analysis section has been rephrased to improve meaning as indicated on page 11.

14. Findings: findings are satisfactorily reported. However, consider indicating the meanings of the various asterisks (* and **) as footnotes for easy reading (provided it is in line with editorial policy of the journal).

Thanks for pointing this out. The asterisks (*, ** & ***) indicate the extent of integration as indicated in the bracketed phrases on pages 10-11.

15. Conclusion and policy implications: Authors should consider including a brief summary of the entire research before highlighting the key findings and some implications for health policy

Thank you for the guidance: A brief summary of the research findings have been included on page 22.

16. Weaknesses: generalizability of the study may not be possible because it was conducted in just one district of the country (Uganda).

This is noted. The generalizability of the study has been inserted as a weakness pages 21-22.

17. Secondly, the authors need to indicate how they controlled for the weaknesses of facilitated panel discussions such as ineffective moderating, dominating panelists, and ill-prepared panelists.

The wording “each panelists was allocated a reference numbers and minimal time as well as equal opportunity to share their views” has been inserted in the methods section on page 9.

18. Finally, please consider proofreading and correcting minor grammatical errors throughout the entire write-up.

Thank you for pointing out this. The typographical error have been addressed throughout the manuscript.

Response to Reviewer two’s Comments

19. It is a well-written manuscript however, we may consider the following to improve the quality of this very informative paper. Background..

Thank you for the complement.

20. (Pg.5, line 119-123) …… There is a need to be very clear on the specific objectives. It appears to me that, there is a repetition (Mentioning health system building blocks and again service delivery; I think service delivery is part of the health system building blocks…and even health facility is a component within service delivery) …… We may consider our specific objective to be 1) To determine the extent of NACS integration across the pillars of the health system building blocks, ii) To determine the extent of NACS integration beyond the health system building blocks (Community health system) ----Detail provided in https://gh.bmj.com/content/3/Suppl_3/e001384 and; 3) To explore facilitators and barriers for NACS integration.

This is noted. Service delivery is part of the health system. The sentences have been merged to avoid repetition and improve meaning on page 6 and now it reads as follows; “Therefore, this study aimed to assess the extent of NACS integration within and beyond the health system and identify key drivers to this process. The study specifically examined the incorporation of NACS within the following domains: 1) healthcare facility levels, 2) health system building blocks and the community system. Thank you for this additional resource that has been used for beefing up the paper.

21. Methodology

In the title and abstract case study is mentioned; but in the methodology part case study is not mentioned. Could you clarify?

The wording “Case study” has been deleted under the abstract, title and methods section. The wording “A case of Tororo district” has been maintained in the title for emphasis. The study employed a mixed method design.

22. Discussion

 It would be better to include the strengths and limitation of the study

Strengths and limitations of this study has been inserted after the discussion section on page 21. 

23. P.g 21(line 469-487), please reorganize your recommendations into practice, policy level, and research (I think there may be some

We agree. The recommendation has been arranged in into practice, policy level, and research as guided on pages 22-23.

---

## [Decision Letter · Decision Letter 1]

30 Oct 2023

Extent of integration of nutrition assessment counselling and support interventions in the health system and respective drivers: A case of Tororo district, Uganda

PONE-D-23-21035R1

Dear Dr. Namukose,

We’re pleased to inform you that your manuscript has been judged scientifically suitable for publication and will be formally accepted for publication once it meets all outstanding technical requirements.

Kind regards,

Imran Naeem

Academic Editor

PLOS ONE

Additional Editor Comments (optional):

Reviewers' comments:

Reviewer's Responses to Questions

**Comments to the Author**

1. If the authors have adequately addressed your comments raised in a previous round of review and you feel that this manuscript is now acceptable for publication, you may indicate that here to bypass the “Comments to the Author” section, enter your conflict of interest statement in the “Confidential to Editor” section, and submit your "Accept" recommendation.

Reviewer #2: All comments have been addressed

2. Is the manuscript technically sound, and do the data support the conclusions?

Reviewer #2: Yes

3. Has the statistical analysis been performed appropriately and rigorously? 

Reviewer #2: Yes

4. Have the authors made all data underlying the findings in their manuscript fully available?

Reviewer #2: Yes

5. Is the manuscript presented in an intelligible fashion and written in standard English?

Reviewer #2: Yes

6. Review Comments to the Author

Reviewer #2: The Authors have addressed my previous comments. The Authors have addressed all the key comments from my side; 1. The wording “Case study” has been deleted under the abstract, title and methods section. 2. Strengths and limitations of this study has been inserted and 3).The recommendations have been arranged as advised

7. PLOS authors have the option to publish the peer review history of their article (what does this mean?). If published, this will include your full peer review and any attached files.

Reviewer #2: **Yes: **James Tumaini Kengia

---

## [Editor Report · Acceptance letter]

13 Dec 2023

PONE-D-23-21035R1 

PLOS ONE

Dear Dr. Namukose, 

I'm pleased to inform you that your manuscript has been deemed suitable for publication in PLOS ONE. Congratulations! Your manuscript is now being handed over to our production team.

Kind regards, 

on behalf of

Dr. Imran Naeem 

Academic Editor

PLOS ONE